# Electrospinning of Potential Medical Devices (Wound Dressings, Tissue Engineering Scaffolds, Face Masks) and Their Regulatory Approach

**DOI:** 10.3390/pharmaceutics15020417

**Published:** 2023-01-26

**Authors:** Luca Éva Uhljar, Rita Ambrus

**Affiliations:** Faculty of Pharmacy, Interdisciplinary Excellence Centre, Institute of Pharmaceutical Technology and Regulatory Affairs, University of Szeged, Eötvös Street 6, H-6720 Szeged, Hungary

**Keywords:** electrospinning techniques, electrospun medical devices, face mask, healthcare, melt electrospinning, regulation, solvent electrospinning, tissue engineering, wound dressing

## Abstract

Electrospinning is the simplest and most widely used technology for producing ultra-thin fibers. During electrospinning, the high voltage causes a thin jet to be launched from the liquid polymer and then deposited onto the grounded collector. Depending on the type of the fluid, solution and melt electrospinning are distinguished. The morphology and physicochemical properties of the produced fibers depend on many factors, which can be categorized into three groups: process parameters, material properties, and ambient parameters. In the biomedical field, electrospun nanofibers have a wide variety of applications ranging from medication delivery systems to tissue engineering scaffolds and soft electronics. Many of these showed promising results for potential use as medical devices in the future. Medical devices are used to cure, prevent, or diagnose diseases without the presence of any active pharmaceutical ingredients. The regulation of conventional medical devices is strict and carefully controlled; however, it is not yet properly defined in the case of nanotechnology-made devices. This review is divided into two parts. The first part provides an overview on electrospinning through several examples, while the second part focuses on developments in the field of electrospun medical devices. Additionally, the relevant regulatory framework is summarized at the end of this paper.

## 1. Introduction

Nanofibers are unidirectionally elongated polymer-based solid structures with a nanosized diameter. The fabrication of nanofibers is possible in numerous ways, e.g., self-assembly, melt blowing, and drawing, although electrospinning is the most common. Electrospun nanofibers have various applications under investigation, such as functional textiles, functional clothing, skincare and cosmetics, electronics, acoustics, composites, filters, and biomedical uses [1]. The latter category includes drug delivery systems, wound dressings, and cell scaffolds. This field of research is very intense, with more and more scientific articles being published every year (Figure 1). The advantages of nanofibers in biomedicine are the wide range of suitable polymers, the possibility of loading and controlled release, as well as the small diameter, the large surface area, and the controllable pore structures. With electrospinning, the 3D structure of the nanofiber mat can be tailored, creating an architecture very similar to the extracellular matrix, which is desirable in chronic wound healing and tissue engineering.

Chronic or non-healing wounds develop when any acute wound fails to heal in the expected time frame for that type of wound, which might be a couple of weeks or up to six weeks. Ulcers are the most common type of chronic wound, which may be caused by constant pressure (decubitus ulcer), venous or arterial circulatory problems, or diabetic angio-neuropathy. These wounds need special care and innovative dressings to promote the healing process [2]. The wound surface is frequently covered by necrotic tissue and bacterial biofilm, necessitating debridement and local disinfection. Electrospun wound dressings not only provide a suitable environment for cell growth, they can also be loaded with antibiotics or disinfectants [3].

Tissue engineering involves the implantation of a natural, semi-synthetic, or synthetic implant to repair damaged tissue. The closer the implant’s properties are to the original tissue, the more successful the operation. As a result, while autograft is the best option in this case, it is not always feasible. Scaffolds primarily comprised of polymeric biomaterials provide structural support for cell adhesion and subsequent tissue formation when synthetic or semi-synthetic implants are required. One suitable way to create differently designed scaffolds is electrospinning [3].

The COVID-19 pandemic spotlighted the importance of personal protective equipment, such as face masks. Face masks proved to reduce the human-to-human transmission of the virus by stopping the spread of virus-containing saliva and respiratory droplets [4]. Typically, these masks have a filter layer, which can be an electrospun nanofiber mat. Several of these masks entered the market in the last few years, e.g., the Bio Hygienic Mask and Inofilter^®^ (Lille, France) 95/99.

The regulation of substances and devices used to maintain and restore health is receiving particular attention worldwide. The two bodies that have a significant influence on other countries’ national agencies are the US Food and Drug Administration (FDA) and the European Medicines Agency (EMA); however, the World Health Organization (WHO) also regularly gives guidelines and publishes various tools for national health systems. Generally, the regulation of conventional medical devices is well-defined, although it is not harmonized between countries [5]. However, regarding nanomedicines and health-related nanomaterials, the number of guidelines is not satisfactory, which can affect the attitude of the industry negatively [5].

The healthcare industry is regulated by strict rules, so considering regulatory requirements at the development stage can be worthwhile. To the best of the authors’ knowledge, there is no published review of the regulatory aspects of nanofibers developed as potential medical devices in the literature. In the first part of this review, the basic principles of electrospinning, the effects of the different parameters on the process and the produced fibers, and the types of solvent and melt electrospinning techniques are discussed. In the second part, the focus is on electrospun medical devices. After a summary of the possible areas of application, their regulatory approach is described.

## 2. Electrospinning

### 2.1. Principles of Electrospinning

The most common production method for nanofibers is electrospinning due to its simplicity and industrial scalability. Electrospinning is an electrohydrodynamic phenomenon in which the fibers are formed by drawing out a polymer jet under a high electric field. To accomplish this, a fluid polymer is required at the starting point, which can be either a polymer solution/emulsion or a melted polymer. Accordingly, solution, emulsion, and melt electrospinning methods can be distinguished. The methods are described in greater depth later.

Generally, electrospinning is driven by the electrostatic potential characterized by high voltage (some tens of kV) and very low current (10^−7^–10^−3^ mA) [6]. The most common way to generate electrostatic potential is to apply a positive voltage to the polymer solution (typically onto the spinneret), while the metal collector is either grounded or negatively charged on the other side (Figure 2A). Nanofibers can also be electrospun with the inverse method when the collector is charged positively (Figure 2B); however, it was found to be an inferior method in terms of both fiber properties and productivity [7]. The potential difference between the charged liquid polymer and collector plays a key role in nanofiber production as it generates the electric field.

In all cases, the initial step in electrospinning is the formation of the Taylor cone. The surface tension always attempts to reduce the surface area of a liquid, the polymer droplet tends to be spherical in the absence of a strong voltage (Figure 3A). However, when a high voltage appears, the electric field acts on the polymer and deforms its shape (Figure 3B). The so-called Taylor cone is an elongated droplet that is formed when the surface tension is balanced by the localized charges generated by the electrostatic force (Figure 3C). The cone is named after Sir Geoffrey Taylor, who first modeled the phenomenon in 1964 [8]. When the electrostatic force can overcome the surface tension, a fine, charged polymer jet is ejected from the tip of the Taylor cone [9,10,11] (Figure 3D). The jet tends to discharge itself, so it travels in the air toward the grounded or oppositely charged collector where the fiber deposition occurs. The sufficient cohesive force that exists in the polymer stabilizes the jet and allows it to elongate, stretch, whip, and thin while traveling.

### 2.2. Effects of Different Electrospinning Parameters

The factors that determine the properties of nanofibers are usually grouped into three categories (Figure 4):Process parameters (high voltage, flow rate, and distance between the Taylor cone and the collector);Material properties (viscosity—related to the molecular weight and concentration of the polymer, surface tension, conductivity, and volatility of the solvent);Ambient parameters (temperature and humidity).

#### 2.2.1. Process Parameters

Among the process parameters, the applied high voltage has the greatest influence on the mechanism of electrospinning and thus the morphology of the formed fibers. Generally, a higher applied voltage stretches the polymer liquid more, resulting in thinner fibers [12,13]. However, Wu et al. described a point where increasing the voltage had the opposite effect, resulting in fibers with a larger diameter [7]. Furthermore, there are even reports showing an increase in fiber diameter with increasing voltage [14,15]. It is also notable that applying too high of a voltage forces the primary jet to emerge into several secondary jets, forming multiple jets that lead to beads and non-uniform fibers [16].

In addition to the high voltage, other factors may influence the electrospinning process. Certainly, it can be influenced by the equipment itself and the process parameters. The equipment can have three different orientations depending on the initial and target positions of the jet. The electrospinning setup can be horizontal or, in the case of a vertical arrangement, top-down or bottom-up. The degree to which the electrical force and gravitational force contribute differs among the different setups. Although this is often overlooked, the gravitational force could influence the shape of the Taylor cone, jet trajectory, fiber diameter, distribution, and overall spinning efficiency according to Suresh et al. [17].

Additionally, to obtain uniform nanofibers on the collector, complete drying of the jet, meaning evaporation of the solvent or solidification of the melted polymer, is required. Therefore, the jet requires an appropriate flight time in the air while whipping toward the collector. With a longer distance, the flight time increases, and thinner fibers will be formed. In this way, the flight distance of the jet (tip-to-collector distance in the case of nozzle electrospinning) affects the fiber morphology [3,18]. However, with the increase of the distance, the strength of the electric field drastically decreases since it is inversely related to the square of the distance. Hence, the proper distance for each electrospinning process depends on the voltage and other parameters, usually ranging from 10 cm to 25 cm.

During production, the electrospun nanofibers are deposited on the collector, forming the nanofiber mat on its surface. The orientation of the fibers is affected by the type of collector. The random fiber orientation is sufficient for the majority of drug delivery applications. Nevertheless, some fields (mainly tissue engineering) require structured scaffolds with aligned fibers [19]. The basic collector is a flat metal plate, often wrapped in aluminum foil, on which the nanofibers are randomly arranged. However, as electrospinning continues, the fiber mat thickens and insulates the collector, causing a decrease in the electric field and incomplete solvent evaporation [20]. To increase the deposition area, a rotating mandrel collector is widely used. Alfaro De Prá et al. compared a rotating drum, a pair of 6 mm grounded copper wires in parallel positions at a distance of 1 cm from each other, and a rotating mandrel with a diameter of 1 mm. With the rotation of the drum, the fibers stretched, aligned, and thinned. In the case of the parallel wires, the fibers formed a thick mat with nanofibers orientated perpendicular to the wires. The mandrel collector enables the production of tubular scaffolds, which might be applied to the engineering of nerves and blood vessels [21]. Another collector type is the liquid bath collector, which can be used to obtain special three-dimensional electrospun fiber mats collected in a non-solvent, mainly water, ethanol, or methanol [22,23].

In nozzle-based electrospinning, the liquid polymer is fed continuously from a syringe through a nozzle into the electric field by a pump. The process is influenced by the feeding rate. If the feeding rate is increased while the voltage remains constant, the diameter of the collected fibers will increase [24,25]. Too high of a feeding rate, on the other hand, results in beaded fibers due to a lack of appropriate drying time before reaching the collector. Zuo et al. measured the size of the beads on the improper fibers and found that if the feeding rate was increased from 2 mL/h to 3.5 mL/h, 5.6 mL/h, and 9 mL/h, holding all other parameters constant, then beads were observed with an average size of 8 μm, 14 μm, and 23 μm, respectively [26].

#### 2.2.2. Material Properties

Among the material properties, the type of polymer affects the charge of the electrospinning liquid. According to the paper of Tong et al., electrospinning of cationic polymers (e.g., chitosan) is only achievable with a positively charged spinneret [12]. The number of charges in the liquid determines its conductivity. Electrospinning cannot take place in polymer solutions with very low conductivity, as there is no charge on the surface of the liquid droplet for the electric field to act on. In this case, neither the Taylor cone nor the jet is formed. On the other hand, solutions with very high conductivity have the disadvantage that they tend to form a multi-jet.

Angammana et al. investigated different poly(ethylene oxide) (PEO) nanofibers by varying conductivity in electrospinning solutions by dissolving sodium chloride. By increasing conductivity, the results showed three phases. In the first phase (0–200 μs/mm), the average jet current increased and the average fiber diameter decreased. However, in the second phase (200–2000 μs/mm), the average jet current slightly decreased, while the fiber diameter kept decreasing. A further increase in conductivity resulted in intermittent or no jet [27].

The polymer’s molecular weight should also be taken into account. In general, a higher molecular weight is preferable because longer polymer chains facilitate the formation of nanofibers by entangling better [28]. Furthermore, the diameter of the fibers is also influenced by molecular weight. Colmenares-Roldán et al. studied the electrospinning of different polycaprolactone solutions and found that the diameter of the fibers decreased with the reduction in the molecular weight of the polymer. Proper fibers were made from 80 kDa and 45 kDa polycaprolactone; however, with 14 kDa, fiber formation was not possible [28]. If higher molecular weight polymers cannot be used in solution electrospinning, the concentration should be increased [11]. Optimizing the concentration of the polymer solution is critical since both too high and too low a concentration will result in imperfect electrospinning [28,29]. Too much of a high molecular weight polymer can get stuck in the nozzle, clogging it and preventing it from spinning, while low concentrations of polymer can break the jets into droplets. As the viscosity of the polymer solution is related to the concentration and molecular weight of the polymer used, both higher molecular weight and higher concentration can lead to the increased viscosity of the fluid [30]. Consequently, the viscosity of the electrospinning solution or the melted polymer should be optimized as well. Feng et al. produced beadless electrospun mats of four different molecular weights of chitosan (50 kDa, 100 kDa, 200 kDa, and 400 kDa) by decreasing the viscosity of the solution by PEO copolymer formation. A certain range of viscosity and polymer concentration could be determined within adequately prepared nanofibers for all molecular weights, and defects occurred above and below this range [31].

The effect of surface tension was also studied in the same article by Feng et al. In their case, it was necessary to reduce the surface tension with the use of ethanol to form beadless fibers of chitosan [31]. In general, water has a high surface tension, so for a stable and continuous jet, a substantially strong electrical force is needed. Therefore, adding surfactants or replacing water may result in smoother and more uniform fibers.

In the case of solvent electrospinning, the type of solvent also influences the structure and properties of the nanofibers formed. Song et al. investigated the effects of different deionized water–ethanol solvent mixtures as solvents of PEO. The results revealed that the type and volatility of the solvent mixture determine many properties of the nanofibers. In brief, with the increase in ethanol, the average fiber diameter significantly increased, the diameter distribution widened, the surface of the nanofibers became rough, and the productivity decreased. Furthermore, as the ethanol content increased, the molecular chain orientation and crystallinity degree decreased [32].

#### 2.2.3. Ambient Parameters

Aside from material attributes and process parameters, ambient parameters such as temperature and relative humidity also affect the electrospinning process and the properties of the obtained nanofibers. The solvent appears to evaporate faster in a warmer environment. Moreover, the environmental temperature affects such properties of the polymer solution or melt as viscosity, surface tension, and conductivity as well [30]. Vrieze et al. demonstrated that temperature has an influence on the morphology of polyvinylpyrrolidone (PVP) nanofibers by both affecting solvent evaporation rate and viscosity. At both low and high temperatures, thinner fibers were more obtainable than at the intermediate temperature. At lower environmental temperatures, the solvent evaporation rate decreased, so the jet elongated more. Meanwhile higher temperatures resulted in lower viscosity, which also facilitated the formation of thinner nanofibers. However, this phenomenon is material-specific so different polymers may act differently [33]. The majority of papers showed a significant indirect influence of temperature on fiber diameter and morphology, concluding that the inverse connection between temperature and viscosity may be exploited to modify nanofibers [34,35].

Relative humidity can have a significant impact on the electrospinning process [19]. At very low humidity, the solvent evaporates rapidly and may cause needle clogging. On the other hand, at extremely high humidity, the polymer jet breaks and electrospraying occurs. Moreover, the relative humidity can affect the fiber morphology, which is related to the hydrophilic or hydrophobic nature of the polymer. In the case of water-soluble, hydrophilic polymers, the fiber diameter can be tailored by varying the humidity. As with low temperature, high humidity slows the evaporation of the solvent, resulting in longer flight times, increased elongation of the jet, and consequently smaller fiber diameters. Additionally, vice versa, lower humidity promotes faster drying, which results in thicker fibers [36,37]. Furthermore, high humidity can block the evaporation of the aqueous solvent, which facilitates the formation of beads. Feng et al. could prepare uniform and beadless chitosan nanofibers with a relative humidity of up to 40%; however, the electrospinning was problematic at higher humidity levels [31].

On the contrary, in the case of hydrophobic polymers, greater humidity leads to the formation of porous nanofibers as a consequence of the complex interaction between water as a non-solvent, the hygroscopic solvent of the polymer, and the polymer itself [38,39].

### 2.3. Electrospinning Methods

As already mentioned above, solution, emulsion, and melt electrospinning can be distinguished. The three methods follow the same main principles, although there are notable differences. Solution and emulsion electrospinning processes are very similar regarding their equipment, advantages, and limitations. The main difference is the type of polymer liquid, as their names imply.

On the other hand, for melt electrospinning, heating is required to melt the polymer. For this reason, an additional heating component is needed, which causes a more complex electrospinning setup. Additionally, both the nanofiber-forming polymer and the active ingredient must be thermally stable, which limits the choice of materials. However, some polymers (e.g., PP or PET) cannot be dissolved, only melted before spinning. Moreover, the absence of solvent makes the melt method environmentally friendly. Finally, melt electrospinning is comparable to additive manufacturing techniques and thus can be utilized to create special three-dimensional electrospun nanofiber scaffolds for regenerative medicine applications [40].

#### 2.3.1. Solution Electrospinning

As mentioned above, solution and melt electrospinning are the two major types of the electrospun nanofiber production process. Despite the fact that more articles on the topic are published each year, the vast majority of them are about solution electrospinning. According to Calori et al., approximately 90% of the ultrafine-based scaffolds were produced by solvent electrospinning in 2020 [41]. The popularity of the method is due to its simplicity, cost-effectiveness, and compatibility with a wide range of polymers. As a drug delivery system, it is advantageous that nanofibers can be loaded not only with small molecules but also with macromolecules or other nanostructures (e.g., liposomes, nanoparticles) [42,43]. The material properties influencing the process can be tailored relatively easily by modifying the solution. The micro- and macrostructure of the resulting nanofiber mat can thus be controlled. In addition, the whole process occurs at room temperature, which protects thermally degradable materials.

##### Nozzle-Based Methods

Generally, the basic solution electrospinning setup contains a high-voltage supply and a polymer container, practically a syringe, a pump, a spinneret (needle or nozzle), and a collector. A single-needle or single-nozzle configuration is the simplest and most common, where the Taylor cone is created on the top of the capillary. It can provide uniform nanofibers from a single-polymer solution or polymer blend [44]. Even if the polymers are not miscible in a common solvent, it is still possible to achieve nanofiber construction from two or more separate solutions that are filled into separate syringes and usually connected to separate pumps, resulting in variable feeding rates. The solutions contact at the tip of the spinneret and form a common Taylor cone. For this, modification of the needle was required, and two distinct configurations, namely side-by-side and coaxial, were developed. The nanofibers produced by the side-by-side configuration show two different sides; hence, they are commonly referred to as Janus fibers [20]. The coaxial spinneret is constructed from two or more concentric needles placed inside each other; thus, the produced nanofibers have a core–shell structure [45]. This structure allows the separate encapsulation of multiple active ingredients, thereby eliminating incompatibility, protecting the active substance from the environment, and controlling the drug release. Furthermore, coaxial electrospinning can be used to produce hollow nanofibers. If the core polymer solution has a very low concentration, then once the solvent evaporates, only a film layer of the polymer remains on the inner surface of the shell polymer, while the rest is an empty core [35]. Additionally, hollow fibers can be obtained by selective removal of the core polymer [46,47]. Hollow nanofibers have a broad range of applications, from drug delivery systems to tissue engineering and optical waveguides [48].

Normally, a single Taylor cone appears at the tip of the spinneret, from which a single jet is ejected. Hence, scaling up the process requires several spinnerets connected in parallel. This so-called multi-nozzle electrospinning method combines the benefits of nozzle-based electrospinning in addition to the ability to produce nanofibers in industrially relevant volumes. Nonetheless, the method is challenging because the presence of several jets at the same time modifies the electric field and the charged jets interact with each other, which increases the difficulty in collecting nanofibers [49]. In this field, Tan et al. recently collected numerous nozzle types and setup designs [20]. Hence, only the categorization is highlighted here to illustrate the variety of the method. Multi-nozzle electrospinning consists of three types: single nozzle with multiple jets, multiple nozzles with a single jet at each nozzle, and multiple nozzles with multiple jets at each nozzle. In addition, there are stationary and rotary techniques, as well as combined methods (e.g., blowing-assisted multi-jet electrospinning).

##### Nozzle-Free Methods

The main drawback of single-nozzle electrospinning is the limited production capacity (0.01–0.1 g/h) [50]. This critical issue prompted intensive research, and in addition to multi-nozzle electrospinning, another possibility seems promising. Nozzle-free methods, also known as free surface electrospinning, offer the potential to produce nanofibers on an industrial scale. In this case, several Taylor cones are formed simultaneously on the surface of the solution, from which nanofibers are drawn. Additionally, by omitting the nozzle, clogging can be avoided, which solves another disadvantage of needle-based electrospinning [49]. However, nozzle-free methods have limitations as well, including the wide range of fiber diameters and the incompatibility with volatile solvents.

Nozzle-free electrospinning can be divided into two groups depending on whether the set-up is equipped with a stationary or a rotating spinneret [20]. Simple polymer solution reservoirs, such as bowls [51], stepped pyramids [52], or plate-edge spinnerets [53], can be used as stationary spinnerets. Moreover, the method can be supplemented with different external forces, such as magnetic force (ferromagnetic liquid electrospinning or magnetic-field-assisted electrospinning), high-pressure gas flow (bubble electrospinning), or acoustic radiation force generated by an ultrasound transducer (ultrasound-enhanced electrospinning) [54]. On the other hand, rotating spinnerets are half immersed in the polymer solution, and the multiple Taylor cones are formed on the surface of the cylinder [55], disc [56], wire [57], or ball [58], while the rotation ensures a continuous solution supply.

#### 2.3.2. Emulsion Electrospinning

During emulsion electrospinning, the homogenous mixtures of two or more immiscible liquids are electrospun. The method is very similar to single-needle solution electrospinning; however, the obtained nanofibers have a core–shell structure [59]. The advantages of the core–shell structure are that it protect the biologically active material located in the core and to ensure its controlled release. In addition, emulsion electrospinning allows the possibility to omit the coaxial needle, which simplifies the production. Nevertheless, it can be considered a green method, since water can be used in the continuous phase and organic solvents can be reduced or avoided [60]. Indeed, the avoidance of toxic solvents is particularly important for nanofibers intended for biomedical use.

#### 2.3.3. Melt Electrospinning

The other large group of electrospinning methods is melt electrospinning, in which the charged jet is drawn out of a polymer melt. The active ingredient and/or excipients can be dissolved in the molten polymer if not only the pure polymer nanofiber is desired. As there is no solvent to evaporate in melt electrospinning, only heat transfer occurs during the travel of the jet. The absence of residual solvent is desirable for some biological applications [61]. Henceforth, solvent-free methods are advantageous in terms of sustainability [62]. Otherwise, melt electrospinning requires the heating of the fiber-forming polymer, which necessitates a more complex apparatus. Additionally, the utilization of nanofibers as drug delivery devices is limited since most active ingredients are not thermally stable at higher temperatures [63]. It should be noted that polymer melts are generally highly viscous; therefore, high extrusion and a sufficiently large electric field are required [59]. Yet, the higher viscosity provides higher stability of the jet compared with the polymer solution jet, which results in better control of the fiber deposition on the collector [64].

The setup for melt electrospinning generally consists of a high-voltage power supply, a polymer container, a heating device, a spinneret, and a collector. The type of melt electrospinning depends on whether the polymer container is a syringe or something else [61].

##### Syringe-Based Method

A plastic or glass syringe reservoir and a metal needle spinneret are major parts of melt electrospinning systems. The main advantages of this setup are the standardized holding capacity and nozzle sizes, the standard polymer delivery, and the interchangeability [61]. The syringe is usually filled with pre-treated, uniformly sized pastilles of the polymer, which are melted in the syringe itself. For melting, various methods are available. The wall of the syringe can be heated by an electric heater or circulating water, but if an even higher temperature is required, so-called heat guns can be used [61,62].

Zhou et al. pointed out that the temperatures at the spinneret and in the spinning region are critical to producing sub-micron-sized fibers. If the environmental temperature was below the glass transition temperature of the polymer, the too fast solidification of the jet led to an increased fiber diameter [65].

##### Syringe-Free Methods

The two main disadvantages of syringe-based melt electrospinning are the potential degradation of the polymer or active ingredient at high temperatures and the larger fiber diameter with less controllable morphology compared with solvent electrospinning [66]. To overcome this, various syringe-free procedures have been developed.

Li et al. reported promising results using laser melt electrospinning, a technique that is similar to commercial fused deposition modeling (FDM). In this setup, a solid polymer rod is fed into the heating zone of the CO_2_ laser. Rapid and uniform laser heating can minimize polymer degradation and facilitate the production of tissue engineering nanofiber scaffolds [67].

There are various articles about reducing fiber diameter, some of which are provided here. Malakhov et al. used a screw extruder instead of a syringe to deliver a constant supply of polymer to the spinneret [68]. Morikawa et al. were able to produce PLC nanofibers with a 5.67 ± 1.51 μm average diameter, which meant a reduction of more than 75% by using a wire electrospinning setup. Instead of a syringe, a Joule-heated wire that is coated with the polymer of choice was used to deliver and melt the PLC [69]. Balogh et al. compared the average fiber diameters of nanofiber mats produced by solvent electrospinning, melt blowing, and syringe-based melt electrospinning and found them increasing in this order. Melt blowing is a gas-assisted method that is composed of a melt extruder and a high-speed gas stream. Except for the spinneret, the setups for melt blowing or melt electrospinning in this investigation (syringe, heater, collector, and other parts) were identical and easily interchangeable [70]. Um et al. took it a step further and used the technique of electroblowing, which combines melt blowing with melt electrospinning. Compared with solution electrospinning, the hot air blown onto the jet improved the spinning of the hyaluronic acid fibers. The nanofiber formation was consistent and uniform at 57 °C blown-air temperature [71]. Furthermore, Zhmayev et al. developed so-called gas-assisted melt electrospinning (GAME), which is similar to electroblowing except that the major attenuation driving force in GAME is the electric field, whereas in electroblowing it is the air. Compared with melt electrospinning, a drastic decrease in fiber diameter was observed [72].

## 3. Biomedical Applications of Electrospun Nanofibers

Electrospun nanofibers have a broad range of potential biomedical applications, including drug delivery systems, wound dressings, tissue engineering scaffolds, various diagnostic tools, in vivo models, and filters, which can be produced via electrospinning. As drug delivery systems, nanofibers can increase the solubility and permeability of BSC II/IV active pharmaceutical ingredients, improve the therapeutic effect, reduce the side effects and toxicity, and facilitate alternative administration. Additional benefits are the tailorable release profile, the high loading capacity, the high encapsulation efficiency, the ease of operation, and the cost-effectiveness [73].

As in wound dressings and tissue engineering, the ability of nanofibers to produce aligned scaffolds capable of mimicking the extracellular matrix is exploited [74,75]. The fiber diameter, the pore size, and the alignment of the nanofibers are important to mimic the nano-sized features of human tissues. These features may play an active role in regulating cell activities, such as orientation, migration, proliferation, and differentiation. Ferraris et al. published an article about nanofiber topography and cell behavior [76].

Nanofibers offer the benefits of flexibility and/or stretchability, conductivity, and transparency, as well as a large surface area and diverse fiber morphology for biosensors and other soft electronics [77]. It is also feasible to develop in vivo models by implanting cells into nanofiber scaffolds, which can then be used, e.g., in oncology research [61]. Finally, the large surface area and small pore size of electrospun nanofiber mats are favorable within different filters, e.g., respiratory mask filters [78].

## 4. Nanofibers as Medical Devices

Indeed, it is useful to distinguish between pharmaceuticals and medical devices when considering the use of nanofibers in biomedicine. The two categories are similar in that both are used to cure, prevent, or diagnose diseases. The difference is that pharmaceuticals contain active substances, which are chemicals in nature and actively interact with the human body. Thus, drug-loaded nanofibers belong to this category. In contrast, medical devices do not contain any active substance, so they do not achieve their purpose through chemical action. The exact definition of medical devices by the WHO is: “An article, instrument, apparatus or machine that is used in the prevention, diagnosis or treatment of illness or disease, or for detecting, measuring, restoring, correcting or modifying the structure or function of the body for some health purpose. Typically, the purpose of a medical device is not achieved by pharmacological, immunological or metabolic means” [79]. According to the WHO, there are approximately 2 million different kinds of medical devices worldwide, with varying degrees of complexity. They range from everyday consumer products, such as wound patches and dentures, to more complex devices, including pacemakers. Based on this, a large number of electrospun nanofibers developed for biomedical use has the potential to become a future medical device. However, classification can be challenging for some nanofiber products (e.g., loaded wound dressings), which is further complicated by the fact that classifications differ across global regulators. Thus, what is deemed a medical device in one country might be considered medicine in another [3].

In addition, medical equipment is a large, special group within medical devices. The WHO definition is the following: “Medical devices requiring calibration, maintenance, repair, user training, and decommissioning—activities usually managed by clinical engineers. Medical equipment is used for the specific purposes of diagnosis and treatment of disease or rehabilitation following disease or injury; it can be used either alone or in combination with any accessory, consumable, or other piece of medical equipment. Medical equipment excludes implantable, disposable, or single-use medical devices”. Electrospun nanofibers provide a wealth of opportunities in the area of soft electronics, such as electrodes, conductors, different kinds of sensors, and batteries. Wang et al. recently released a comprehensive review of this field [77]. This article focuses on medical devices that do not meet the requirements of medical equipment, i.e., wound dressings, tissue engineering scaffolds, and respiratory masks. Table 1 summarizes some recent scientific results on nanofibers that might become medical devices.

The COVID-19 pandemic has recently led to an increased need for face masks, which has boosted research as well. Because the face mask serves as a barrier, the vast surface area and small pore size of electrospun nanofiber mats are preferable. Leung and Sun charged the produced nanofiber filter after electrospinning due to the hypothesis that the negatively charged coronavirus adheres more strongly to positively charged nanofibers. Even at an ultralow pressure drop, the produced filters attained 90% efficiency [133].

For chronic wound management, nanofiber dressings have great potential as they provide most of the properties of the ideal dressing, such as protection against bacteria and external aggression, absorption of excess exudates, adequate gas exchange, providing a moist environment, being painless for the patient, and easily removable. Moreover, electrospun nanofibers can mimic the extracellular matrix; regulate skin cell responses, including proliferation, migration, and differentiation; and thus reduce wound healing time radically; therefore, chronic wounds that are not-healing (e.g., diabetic ulcer) can be closed. Wound healing nanofibers are the subject of extensive research, and the results have been provided in various review publications [135,136,137,138].

The scientific debate over tissue engineering nanofibers is also heated. Figure 5 presents an up-to-date graph of the proportion of articles related to certain tissues or organs. Although the number of articles has multiplied, the percentage distribution remained very similar to the end of 2017, as presented by Maurmann et al. [139]. This means that all areas have continued to be vigorously researched over the past 5 years. Tissue engineering is a quite broad topic, so for those who wish to delve deeper, it is recommended to have a closer look at each area separately. The following review articles have recently been published about bone [140,141,142], vascular [143], neural [144], cartilage [145], cardiac [146], and urologic [147] tissues.

In general, nanofiber scaffolds have several beneficial properties to provide adequate tissue replacement, such as high porosity, large surface area, biodegradability, biocompatibility, and tailorable 3D architecture. They can provide excellent support for cell adhesion, proliferation, and differentiation due to their ability to mimic the required extracellular matrix in biological and mechanical features, such as alignment, nano-topography, stiffness, and tensile strength. By choosing the right method and collector, similar structures to the original tissue can be achieved. With the use of co-electrospinning, coaxial electrospinning, or other switched techniques, the structure possibilities are even wider. Wakuda et al. used coaxial electrospinning to produce non-water-soluble collagen hydrogel nanofibers without using any cross-linkers. The electrospinning resulted in collagen core and PVP shell fibers, which were immersed in ethanol to wash away the PVP shell and gel the collagen. They obtained promising results using human umbilical vein endothelial cells as a potential vascular scaffold [124]. Wang et al. developed a hierarchical scaffold for bone tissue engineering, where the bottom layer was a random gelatin mat, on top of which PCL fibers were built utilizing a melt electrospinning writing technique. PCL microfibers could guide cell orientation, while the gelatin nanofibers promoted cell adhesion and proliferation [116].

In addition to developing new implantable scaffolds, electrospinning is a suitable technique to innovate existing medical devices, e.g., to coat vascular stents or orthopedic implants. The aim could be to prevent neointimal hyperplasia through the local delivery of selective pleiotropic drugs [129].

## 5. Regulatory Aspects

As previously stated in paragraph 5, pharmaceuticals and medical devices are distinguished by the presence or absence of pharmacological effects on the human body. According to the Global Harmonization Task Force’s definition, which is used by the WHO, a medical device, “does not achieve its primary intended action by pharmacological, immunological or metabolic means, in or on the human body, but which may be assisted in its intended function by such means” [79]. However, it is highlighted in the definition that the tissue-containing devices may be medical devices according to some jurisdictions but not according to others, which confirms the observation of Foulkes et al. that the classification is not consistent at the global level [3]. Additionally, EMA has created a group dedicated to “borderline products” and left the task of classification to the national competent authorities [148]. In this paper, the regulatory approaches in the USA and the EU are overviewed.

All medical devices in the USA are regulated by the FDA under the Center for Devices and Radiological Health (CDRH). It differentiates three regulatory classes based on the level of control required to assure the safety and efficacy of the device [149]. Class I includes devices with the lowest risk and is regulated only by general control. In the case of Class II medical devices, special controls combined with general controls are necessary. Class III devices are those that support or sustain human life, so they have the highest risk level and require premarket approval from the FDA to obtain marketing. The premarket approval includes a non-clinical laboratory studies section and a clinical investigations section [150]. Similarly to medicines, the clinical evaluation for Class III devices, such as implantable or other high-risk devices, must be based on evidence gathered through clinical investigation. Moreover, clinical investigations must fulfill the requirements of Good Clinical Practice (GCP) regarding both data quality and integrity and ethical standards [151].

The regulatory process for medical devices in the EU is at the member-state level; however, the EMA is also involved. For example, drug-eluting stents belong to the group “medical devices with an ancillary medicinal substance”, so the EMA plays a role in their assessment—they must meet the requirements of the medical device legislation and be CE marked [148]. The manufacturer can place the CE (Conformité Européenne) mark on a medical device if it meets the safety, health, and environmental protection criteria of the EU, and has also passed a conformity assessment.

The classification of medical devices in the EU is according to the Regulation (EU) 2017/745 and contains four levels: Class I, Class IIa, Class IIb, and Class III [152]. The risk is increasing from Class I to III. Class IIb refers to surgically invasive or active devices, which are partially or completely implanted into the body. The regulation states that all implantable devices and Class III devices must undergo a clinical investigation that follows GCP. 

Furthermore, Special Rule 19 applies to all nanomaterial-associated devices; however, they belong to Class IIa, IIb, or III according to the potential for internal exposure [152]. If the risk of internal exposure is higher, the device is placed in a higher class. Accordingly, it is necessary to evaluate nanofibrous medical devices from this point of view as well. If the electrospun device becomes Class 3, it must undergo a clinical investigation.

Clinical investigation of a medical device can be any systematic investigation involving one or more human subjects, undertaken to assess the safety or performance of a device. It is regulated by the ISO 14155:2020 (clinical investigation of medical devices for human subjects—good clinical practice) standard [153]. In general, clinical investigations require clinical-grade material and an authorized manufacturing site to produce it. ISO 13485 (Medical Devices—Quality Management Systems—Requirements for Regulatory Purposes) is an internationally agreed standard that helps the medical device industries to fulfill the requirements of quality management systems accepted by the regulatory authorities. The ISO 13485:2016 standard is the only quality management system standard in the EU list of harmonized standards, so most manufacturers attempt to obtain the ISO 13485 certificate.

Electrospun nanofibers are nanomaterials because of their nano-sized diameter. At present, the regulation of nanomedicine is not clear, and the number of specific regulatory guidance documents is poor. Different definitions of nanomaterials have been made by different bodies, such as the US National Institute of Health, the European Science Foundation, and the European Technology Platform. Despite that, over 50 nanomedicines (mainly anti-cancer medicines) have been approved and are currently available on the market [3]. Recently, in April 2022, the FDA published guidance titled “Drug Products, Including Biological Products, that Contain Nanomaterials—Guidance for Industry”. It defines nanomaterials, the absence of which has previously been criticized. However, it does not deal with medical devices.

Specifically related to medical devices, the European Commission published the “Guidance on the Determination of Potential Health Effects of Nanomaterials Used in Medical Devices” in 2015. The guidance draws attention to the use of the ISO 10993-1 (Biological Evaluation of Medical Devices—Part 1: Evaluation and Testing Within a Risk Management Process) standard. Moreover, this guidance describes examples that could be produced by electrospinning, namely:Free nanomaterials added to a medical device (e.g., nano-silver in wound dressings);Fixed nanomaterials form a coating on implants to increase biocompatibility (e.g., nano-hydroxyapatite) or to prevent infection (e.g., nano-silver);Embedded nanomaterials to strengthen biomaterials (e.g., carbon nanotubes in a catheter wall) [154].

Face masks have recently received increased attention due to the COVID-19 pandemic. In March 2020, the FDA issued emergency use authorizations for personal respiratory protective devices [155]. There are two groups of face masks: surgical face masks and respirator masks. The former is classified as a Class I medical device, while the latter is considered personal protective equipment by both the FDA and EMA. In both cases, the masks must satisfy certain tests, such as the particle filtration efficiency test. Detailed information on mask types, structure, testing standards, and the types of masks recommended in each situation can be found in the review published by Naragund and Panda [156].

Concerning nanofibrous medical devices, some early birds are on the market or in the pipeline (Table 2).

When developing nanofibers as a potential medical device, there are a few things that should be considered in the early stages of research that could be important from a regulatory perspective. First, the materials used must satisfy the safety requirements. It is recommended to choose a polymer that is considered safe and has been approved by the authorities. Biocompatible and biodegradable polymers are generally preferred. A good strategy could be the innovation of an already authorized device by electrospinning. Campbell et al. made nanofibers from an FDA-approved cyanoacrylate polymer for closing endonasal surgical defects and compared them with Adherus^®^, an FDA-approved common dural sealant [112]. Second, attention should be given to the toxic residue of the solvent used in the electrospinning process. It is advisable to analyze the residual solvent content and, if necessary, execute post-drying. The use of non-toxic solvents is preferable to aggressive and toxic ones, such as chloroform and HFIP. The latter also can be used if the residual solvent is proven to be below the level of acceptance. Another option can be melt electrospinning, since the solvent is not used in this technique. Third, both the electrospinning process and the equipment itself must be suitable for precisely controllable and reproducible production, which can be demonstrated by validation. In this regard, nozzle-based electrospinning is better than nozzle-free because, in the latter, simultaneous jets lead to a non-uniform fiber diameter [29]. Moreover, cellular and in vivo experiments may require nanofibers produced in a cleanroom environment. Finally, following the encouraging in vitro results, in vivo animal studies are crucial, especially for wound dressings and tissue scaffolds, where nanofibers will interact with living cells. Fortin et al. published promising in vitro results followed by negative in vivo results with electrospun tubular vascular conduits. In vitro, the conduits significantly reduced protein absorption and enhanced the adhesion, proliferation, and retention of endothelial cells seeded on the surface. Therefore, an end-to-end common carotid bypass was performed in 10 sheep, although there was no improvement in endothelialization compared with the controls [157].

## 6. Conclusions

Electrospinning is the easiest and most common method for nanofiber fabrication. The number of polymers that can be used is extensive; however, the material properties have a significant impact on the fibers produced, e.g., the viscosity of the liquid polymer is crucial. Additionally, process and ambient parameters affect the electrospinning process and the properties of the fibers. Voltage is considered to be the most influential factor; however, it should be pointed out that the type of collector determines the arrangement of the fibers within the mat.

Electrospun nanofibers have unique physical characteristics that make them suitable for the development of various medical devices, such as dressings for chronic wounds, tissue engineering scaffolds, filters, and soft electronics. Each of these requires different morphologies, physicochemical properties, and architectures of nanofiber mats, all of which can be achieved using various electrospinning techniques. The two major types are solution and melt electrospinning. The previous can be subdivided according to the presence and type of the nozzle. Various nozzle-free techniques have been developed to increase productivity and prevent needle clogging; however, they have not overtaken the popularity of nozzle-based techniques. Side-by-side and coaxial nozzles can provide the controlled location of different polymers within the fiber, creating Janus and core–shell nanofibers, respectively. Melt electrospinning can be subdivided into syringe-based and syringe-free methods and has two main advantages. First of all, it is considered a green technique due to the absence of solvent, which is beneficial in the production of medical devices. Secondly, melt electrospinning drawing enables the building of mats with more complex structures, which can be applied to the fabrication of tissue regeneration scaffolds.

The potential biomedical applications of electrospun nanofibers differ from drug delivery systems to various medical devices, such as filters, soft electronics, tissue engineering scaffolds, and wound dressings. Generally, the large surface area, the tailorable morphology, and the large range of polymers that can be used are the common advantages. Moreover, nanofibers have additional benefits in every field of application, for example, small pore size and flexibility as face masks or biocompatibility and similarity to the extracellular matrix as wound care and implantable devices. In the field of tissue engineering, the most investigated tissues and organs are the bones, the blood vessels, the skin, and other soft tissues (muscle, tendon, valve), but the research is also intensive on cardiac, neural, and cartilage replacements. The challenges and future direction of electrospinning-based biomedical scaffolds are the development of the capability for reproducible, industrial-scale production and extensive pre-clinical and clinical testing before commercialization [4].

In the case of nanofibrous medical devices, basic research is very intensive, but only a few products have received marketing approval so far. Despite the well-defined regulatory approach of medical devices in general, the regulation of nanomaterial-containing ones is unclear. It is agreed that caution is necessary and that nanomaterial-containing medical devices should be classified into higher-risk categories. The guidelines published by the EU and the relevant ISO standards are certainly good cornerstones, and on specific issues, consultation with the authorities might also help. After classification, the rules applicable to the class must be followed.

## Figures and Tables

**Figure 1 pharmaceutics-15-00417-f001:**
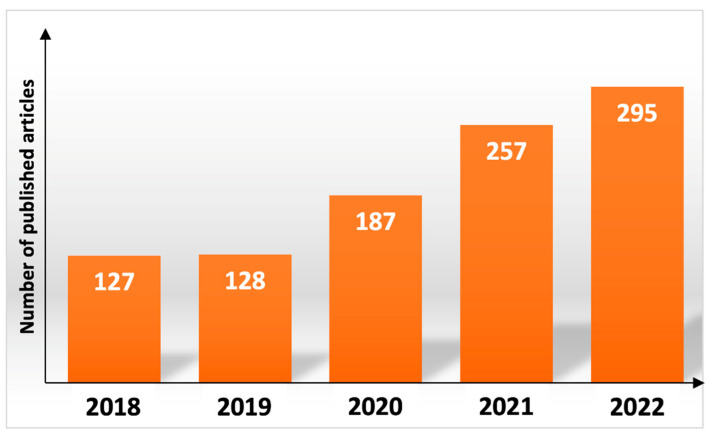
Number of research and review papers published in the previous 5 years according to PubMed database. The keywords used in the field Title/Abstract were: “electrospinning” or “electrospun”.

**Figure 2 pharmaceutics-15-00417-f002:**
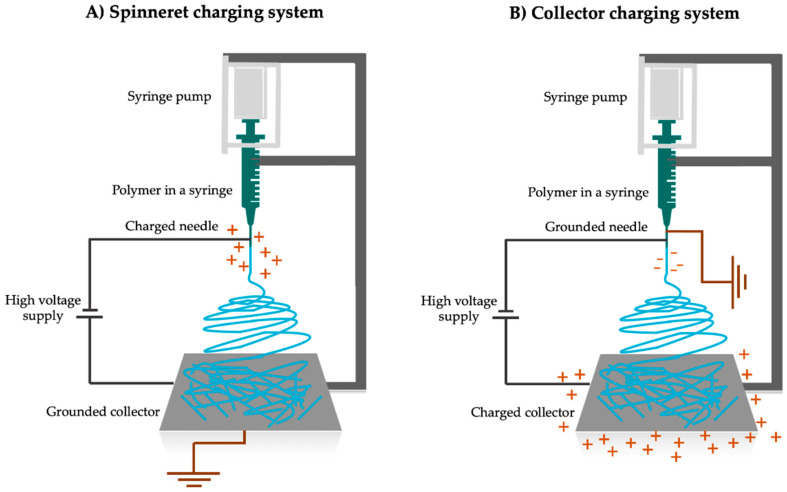
Types of charging systems in electrospinning. (**A**) In a spinneret charging system, a positive voltage is applied to the needle while the collector is grounded. (**B**) In a collector charging system, the needle is grounded while a positive voltage is applied to the collector.

**Figure 3 pharmaceutics-15-00417-f003:**
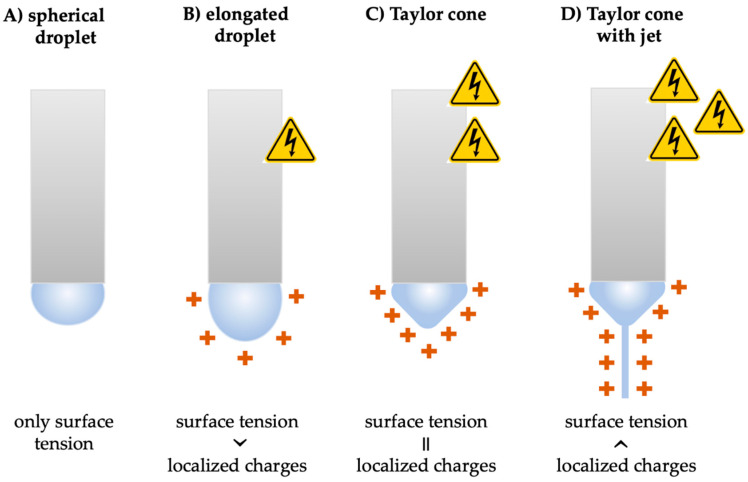
Schematic illustration of the Taylor cone formation as the high voltage increases. (**A**) Without the electric field, the pendant droplet is spherical in shape. (**B**) Localized charges are induced by the electrical force and elongate the droplet. (**C**) The formation of the Taylor cone occurs when the surface tension is balanced by the localized charges. (**D**) Ejection of the charged polymer jet from the Taylor cone due to the additional increase of the high voltage.

**Figure 4 pharmaceutics-15-00417-f004:**
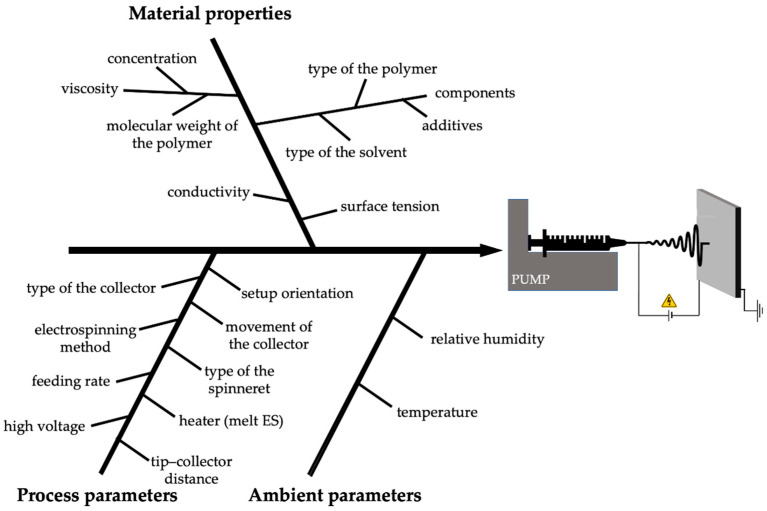
The most important parameters affecting the feasibility of nanofiber production by the electrospinning method.

**Figure 5 pharmaceutics-15-00417-f005:**
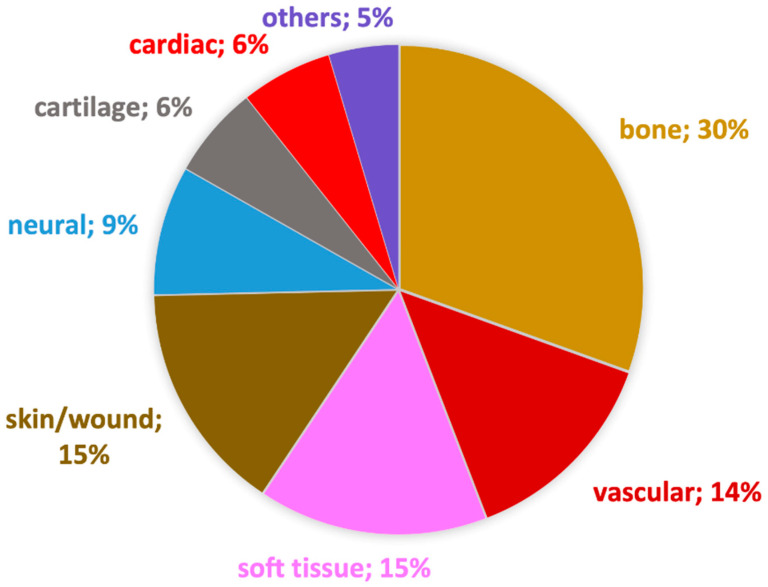
Major application areas of electrospun nanofibers developed for tissue engineering according to research realized in the PubMed database, until October 2022. The keywords used in the field Title/Abstract were: “tissue engineering” and “electrospinning” or “electrospun” and the following tissues or organs: “bone”, “vascular” or “vessels”, “soft tissue” or “tendon” or “valve” or “muscle”, “skin” or “wound healing” or “wound dressing”, “neural” or “nervous”, “cartilage” or “trachea”, “heart” or “cardiac”, “suture”, “bladder”, “corneal”, “liver” or “hepatic”, “incontinence”, “conjunctival”.

**Table 1 pharmaceutics-15-00417-t001:** Recent studies on electrospun nanofibers as potential medical devices.

Type of Use	Type of Production	Polymer	Solvent	Bioactive Agent	Excipients	Cell Line/Animal	Ref.
Wound dressing	Single-needle solution ES	Starch-TPU	DMSO, DMF	-	-	NHDFs, Sprague Dawley rats	[80]
Wound dressing	Single-needle solution ES	Functionalized CS	80% AcOH	-	PEO (sacrificial polymer), 2-formylphenyl-boronic acid (imination reactant/reagent)	NHDFs	[81]
Wound dressing	Single-needle solution ES	Alginate/dextran and alginate/PEO	DW, pH 5.5 PBS	-	PBS (conductivity); P407 and TX100 (surface tension)	NHDFs	[82]
Wound dressing	Single-needle solution ES	PVDF, zP(S-r-4VP) zwitterionic copolymer	DMF, acetone	-	-	L929 mouse fibroblast cells, mice	[83]
Wound dressing	Single-needle solution ES	SF/PCL	HFIP	-	CS and COL type I (coat)	NHDFs	[84]
Wound dressing	Coaxial ES	DMF, DW	-	-		L929 mouse fibroblast cells, Albino Wister rats	[85]
Wound dressing	Single-needle solution ES	CS/PEO	80% AcOH	-	HA (coat)	NHDFs	[86]
Wound dressing	Single-needle solution ES	CS/PEO	80% AcOH	-	-	NHDFs	[87]
Wound dressing	Single-needle solution ES	GEL/PCL	HFIP	Ag and Mg ions	-	NHDFs, HUVECs, Sprague Dawley rats	[88]
Wound dressing	Single-needle solution ES	PVA/CS/starch	DW, AcOH	-	-	L929 mouse fibroblast cells	[89]
Wound dressing	Single-needle solution ES	CS/PCL—HA/PEO bilayered scaffolds	Formic acid, acetone, DW	-	-	Vero cell (monkey epithelial cell line)	[90]
Wound dressing	Single-needle solution ES	CS/PEO	AcOH, DW	-	Benzophenone (photoinitiator for photo-crosslinking)	-	[91]
Wound dressing	Single-needle solution ES	PCL/COL	HFIP	-	-	Human foreskin fibroblasts, Sprague Dawley rats	[92]
Wound dressing	Single-needle solution ES	CS/PVA	80% AcOH, DW	-	NaBH4 solution (3D layered NF sponge creation)	3T3 mouse fibroblasts, JB6 epidermal cells, C57BL/6 mice	[93]
Wound dressing	Single-needle solution ES	VDF-TeFE/PVP	Acetone, isopropanol, DMF	ZnO	-	Wistar rats	[94]
Wound dressing or implant coating	Single-needle solution ES	PET	TFA, DCM	Ag nanoparticles	-	AFSCs, CD1 mice	[95]
Skin tissue scaffold	Co-ES + electroblowing	Soy protein isolate/PEO—PEO	HFIP, ethanol	-	-	HDFBs, RAW 264.7 murine macrophage cell line	[96]
Skin tissue scaffold	Hierarchical construction ES (Sandwich mode)	PLGA/SF	THF, DMF, formic acid	-	-	Human skin stem cells	[97]
Skin tissue scaffold	Single-needle solution ES	PCL/silk sericin	TFE, formic acid	-	3D-printed CS/sodium alginate hydrogel (bottom layer)	NHDFs	[98]
Skin tissue scaffold	Single-needle solution ES	PCL	DCM, DMF	-	Poloxamer 407	BMSCs (C57BL/6 mice)	[99]
Tissue engineering scaffolds	Single-needle solution ES	EPU/SF	TFA	-	-	Fibroblast cells from human neonatal foreskin	[100]
Tissue engineering scaffolds	Single-needle solution ES	PLA/CS	Chloroform, AcOH	-	-	GM07492 human fibroblast cells	[101]
Tissue engineering scaffolds	Single-needle solution ES	PCL	HFIP	-	Neutralized COL (coat)	C57BL/6 mice, de-identified healthy small intestine tissues from discarded surgical samples of infant, teenager or adult	[102]
Tissue engineering scaffolds	Wet ES (+CO_2_ foaming)	PLA	Chloroform, DMF	-	-	NIH 3T3 fibroblasts	[103]
Tissue engineering scaffolds	Coaxial ES	PU/CS	THF, DMF	-	PEO (co-spinning polymer of CS)	-	[104]
Tissue engineering scaffolds	Single-needle solution ES	PCL	Acetone	Y_2_O_3_ nanoparticles	-	L-929 mouse fibroblast cells, UMR-106 rat osteoblast-like cells, Sprague Dawley rats	[105]
Tissue engineering scaffolds	Single-needle solution ES	PCL	DCM, DMF	-	-	Green fluorescent protein (GFP)-labeled fibroblasts, rat neural progenitor cells, rats	[106]
Tissue engineering scaffolds	Coaxial ES	PCL—core; PEG-NB—shell	HFIP	-	Irgacure 2959 (photoinitiator for UV polymerization)	Bovine pulmonary artery endothelial cells, Sprague Dawley rats	[107]
Tissue engineering scaffolds	Single-nozzle solution ES combined with extrusion-based 3D-printing technology	PS	DMF, THF	-	85% phosphoric acid solution (doping agent)	-	[108]
Tissue engineering scaffolds	Single-needle solution ES	PU/carbon nanotube composites	DMF	_	_	HUVECs	[109]
Bladder tissue engineering scaffolds	Coaxial ES	PLCL—core; HA—shell	HFIP, formic acid	-	-	Rat bladder smooth muscle cells, Sprague Dawley rats	[110]
Bladder tissue engineering scaffolds	Single-needle solution ES	PLCL	HFIP	-	COL type I (coat)	hADSCs, Sprague–Dawley rats	[111]
Dura mater substitute	Near-field solution ES	*n*-octyl-2-cyanoacrylate	-	-	-	Harvested dura	[112]
Dura mater substitute	Coaxial ES	Tetramethylpyrazine—core; PLGA—shell	Ethanol, HFIP	-	CS (PLGA/CS graft)	SH-SY5Y human neuroblastoma cells, fibroblasts	[113]
Dura mater substitute (triple-layered)	Single-needle solution ES—inner and middle layer; melt-based electrohydrodynamic printing—outer layer	PCL	HFIP	Gentamicin—inner layer; nano-hydroxyapatite—outer layer	-	NHDFs, MC3T3-E1 cells	[114]
Interface tissue engineering scaffolds	Single-needle solution ES	PCL	Chloroform, DMF	-	-	-	[115]
Oral hard- and soft-tissue engineering scaffolds	Melt ES writing	PCL	-	-	-	MG63 human osteoblast-like cells, HaCaT keratinocyte cells, L929 fibroblast cells	[40]
Bone tissue engineering scaffolds	Single-needle solution ES + melt ES writing	GEL—solution ES; PCL—melt ES writing	AcOH	-	-	Saos-2 cells	[116]
Bone tissue engineering scaffolds	Modified free surface (bubble) ES	PVA	DW	-	Sodium dodecyl benzene sulfonates (surfactant)	-	[117]
Bone tissue engineering scaffolds	Single-needle solution ES	HA/PEO, PVA	DW	TGF-β 2, Baicalein	-	-	[118]
Bone tissue engineering scaffolds	Single-needle solution ES	CA/PCL	HFIP	-	CS (aerogel)	MC3T3-E1 murine osteoblast cells	[119]
Artificial blood vessels	Single-needle solution ES	dPCU	HFIP	-	-	Sprague Dawley rats	[120]
Artificial blood vessels	Multi-nozzle solution ES and co-ES	Bovine GEL/PCL	20% AcOH, DMF, DCM	-	-	3T3 mouse fibroblasts	[121]
Artificial blood vessels	Single-needle solution ES—inner layer; co-ES—outer layer	RHC/PCL—inner layer; PCL—outer layer	HFIP, ethanol	-	PEO (sacrificial polymer)	HUVECs—inner layer; A7r5 rat smooth muscle cells—outer layer	[122]
Artificial blood vessels	Single-needle solution ES	PEUU	HFIP	Heparin	PEG (to earn PEUU@PEG-Hep grafts)	HUVECs, rats and New Zealand white rabbits	[123]
Artificial blood vessels	Coaxial ES	COL	DW	-	PVP (sacrificial polymer)	HUVECs	[124]
Artificial blood vessels	Single-needle solution ES + magnetic environment—inner layer; double-nozzle ES—middle layer; single-needle solution ES—outer layer	PLCL/COL–PLGA/SF–PLCL/COL tri-layer graft	HFIP	-	-	HUVECs, smooth muscle cells, male nude mice	[125]
Cardiovascular stent coating	Coaxial ES	PU—core; PECA—shell	THF, DMF, acetone, DMSO	-	-	NIH-3T3 mouse fibroblasts, platelet	[126]
Cardiovascular stent coating	Single-needle solution ES	Co-recombiner silk-elastin	TFE	-	-	HUVECs	[127]
Drug-eluting stent coating	Single-needle solution ES	PCL/HSA	HFIP	Paclitaxel	Triethylamine	Rabbit iliac artery (drug-release study)	[128]
Drug-eluting stent coating	Single-needle solution ES	CS/PEO/HPβCD	90% AcOH	Simvastatin	-	HPMEC	[129]
Drug-eluting stent coating	Single-needle solution ES	PLGA	HFIP	Vildagliptin	-	HUVECs, New Zealand white rabbits	[130]
Drug-eluting stent coating	Microfluidic ES	GelMA/PEGDA—inner layer; PCL—outer layer	DW, methanol, DCM	Heparin, VEGF	Polydopamine (adherence enhancer), 2-hydroxy-2-methylpropiophenone (photoinitiator for photocrosslinking)	HUVECs, HUASMCs, New Zealand white rabbits	[131]
Respiratory mask	Nozzle-free ES (NTP120 setup)	PAN	DMF	Tea tree essential oil	Polyamidoamine dendritic polymers (drug delivery)	-	[132]
Respiratory mask	Corona ES	PVDF	DMF, acetone	-	-	-	[133]
Respiratory mask	Single-needle solution ES	PCL	Acetone	-	-	-	[134]

Abbreviations: AcOH—acetic acid, AFSC—human amniotic fluid stem cells, BMSCs—bone marrow mesenchymal stem cells, CA—cellulose acetate, COL—collagen, CS—chitosan, DCM—dichloromethane, DMF—*N*,*N*-dimethylformamide, DMSO—dimethyl sulfoxide, dPCU—degradable polycarbonate urethane, DW—deionized water, EPU—elastomeric polyurethane, ES—electrospinning, GEL—gelatin, GelMA—methylacrylated gelatin, HA—hyaluronic acid, hADSCs—human adipose-derived stem cells, HAS—human serum albumin, HDFBs—primary human dermal fibroblasts, HFIP—1,1,1,3,3,3-hexafluoro-2-propanol, HPβCD—2-hydroxypropyl-beta-cyclodextrin, HPMEC—human pulmonary microvascular endothelial cells, HUASMCs—human umbilical arterial smooth muscle cells, HUVECs—human umbilical vein endothelial cells, NF—nanofiber, NHDFs—normal human dermal fibroblasts, P407—Kolliphor P407 poloxamer, PAN—polyacrylonitrile, PBS—phosphate-buffered solution, PCL—polycaprolactone, PECA—Poly(ethyl2-cyanoacrylate), PEG-NB—polyethylene glycol norbornene, PEGDA—polyethylene glycol diacrylate, PEO—poly(ethylene oxide), PET—polyethylene terephthalate, PEUU—poly(ester urethane)urea, PLA—polylactic acid, PLCL—poly(l-lactide)/poly(e-caprolactone), PLGA—poly(lactic-co-glycolic acid), PS—polystyrene, PU—polyurethane, PVA—poly(vinyl alcohol), PVDF—poly(vinylidene fluoride), PVP—polyvinylpyrrolidone, RHC—recombinant human collagen peptides, Saos-2—human osteosarcoma cells, SF—silk fibroin, TFA—trifluoroacetic acid, TFE—tetrafluoroethylene, THF—tetrahydrofuran, TPU—thermoplastic polyurethane, TX100—Triton X-100, VDF-TeFE—vinylidene fluoride-tetrafluoroethylene copolymer, VEGF—vascular endothelial growth factor, Y_2_O_3_—Yttrium oxide, ZnO—zinc oxide.

**Table 2 pharmaceutics-15-00417-t002:** Examples of commercially available electrospun medical devices.

Brand Name	Intended Use	Approved
Bio Hygienic Mask	Compostable mask with FFP2-like filtration capacity	Spain
Bioweb™	Stent coating composite	In the pipeline
Cerafix^®^ Dura Substitute	Regenerative dural repair patch	USA
Covora™	Soft-tissue engineering matrix	USA
EktoTherix™	Soft-tissue scaffold	Completed clinical trial
Inofilter^®^ 95/99	Face mask	USA
PK Papyrus	Covered stent	USA
ReBOSSIS-J	Absorbent bone regenerated material	Japan
ReDura™	Regenerative dural repair patch	Unknown status clinical trial
Restrata^®^ Wound Matrix	Absorbable wound dressing	USA
Rivelin^®^ plain patches	Wound patches	Completed clinical trial

## Data Availability

Not applicable.

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
