# Peer review of "Electrospinning of Potential Medical Devices (Wound Dressings, Tissue Engineering Scaffolds, Face Masks) and Their Regulatory Approach"

_pharmaceutics, 2023, doi:10.3390/pharmaceutics15020417_

Round 1
Reviewer 1 Report (Previous Reviewer 2)
the authors have corrected and attended to my comments. In my point of view is ready for publication
Author Response
Thank you very much for your previous suggestions and recommending the approval of the manuscript.
Best regards,
- Luca Éva Uhljar
prof. Rita Ambrus
- 01. 2023. Szeged, Hungary

Reviewer 2 Report (New Reviewer)
The review by Luca Éva Uhljar et al. gives an overview on the advances in electrospinning for medical devices application. Some minor revisions should be performed before publication:
1. The authors are recommended to give the number of papers and patents (a graph could be added) in this field in the introduction in order to justify a review in this topic. The keywords used to make this graph should be added in the figure legend.
2. Some recent reviews in the area of electrospinning for biomedical application like 10.1016/j.mtchem.2022.100944, 10.1021/acs.chemrev.8b00593, 10.1016/j.apmt.2022.101473 should be mentioned in the introduction. A justification for an additional review in this field should be given.
3. Some tables or figures are recommended to be added to summarize the electrospinning-related parameters, which could make the manuscript more readable.
4. Some statements feel they are lacking references. Admittedly, some of these statement might be considered well known facts, the concepts mentioned might have been referenced previously or will be in the future, but it might still be pertinent add references next to these statements for new readers to the field, skim readers or people who don't necessarily want to go looking for the relevant reference.
5. The authors should make a critical review instead of plain text flow. This means that a comparative discussion should take place assisted with categorization of the attributes of the different systems in each section/Tables.
6. As well known that, electrospinning-based strategies have been widely investigated for biomedical application. Do they have any commercial products based on nanomaterial technique available now?
7. The challenges and future direction of electrospinning-based biomedical scaffolds should be discussed in the Conclusions section.
8. The paper contains some typo and graphical errors. Please read carefully and correct Them.
Author Response
The review by Luca Éva Uhljar et al. gives an overview on the advances in electrospinning for medical devices application. Some minor revisions should be performed before publication:
- The authors are recommended to give the number of papers and patents (a graph could be added) in this field in the introduction in order to justify a review in this topic. The keywords used to make this graph should be added in the figure legend.
Thank you for the suggestion. Accordingly, Figure 1 has been placed in the manuscript.
Electrospun nanofibers have various applications under investigation, such as functional textiles, functional clothing, skincare and cosmetics, electronics, acoustics, composites, filters, and biomedical uses [1]. The latter category includes drug delivery systems, wound dressings, and cell scaffolds. This field of research is very intense, with more and more scientific articles being published every year (Figure 1).
Figure 1. Number of research and review papers published in the previous 5 years according to PubMed database. The keywords used in the field Title/Abstract were: “electrospinning” or “electrospun”
- Some recent reviews in the area of electrospinning for biomedical application like 10.1016/j.mtchem.2022.100944, 10.1021/acs.chemrev.8b00593, 10.1016/j.apmt.2022.101473 should be mentioned in the introduction. A justification for an additional review in this field should be given.
Thank you. The Introduction have been improved with the suggested references and a new paragraph about the face masks.
The wound surface is frequently covered by necrotic tissue and bacterial biofilm, necessitating debridement and local disinfection. Electrospun wound dressings not only provide a suitable environment for cell growth but they can also be loaded with antibiotics or disinfectants [3].
Tissue engineering involves the implantation of a natural, semi-synthetic, or synthetic implant to repair damaged tissue. The closer the implant's properties are to the original tissue, the more successful the operation. As a result, while autograft is the best option in this case, it is not always feasible. Scaffolds primarily comprised of polymeric biomaterials provide structural support for cell adhesion and subsequent tissue formation when synthetic or semi-synthetic implants are required. One suitable way to create differently designed scaffolds is electrospinning [3].
Covid-19 pandemic spotlighted the importance of the personal protective equipment, such as face masks. Face masks proved to reduce the human-to-human transmission of the virus by stopping the spread of virus-containing saliva and respiratory droplets [4]. Typically, these masks have a filter layer, which can be an electrospun nanofiber mat. Several of these masks entered the market in the last few years, e.g.: Bio Hygienic Mask and Inofilter® 95/99.
The novelty of the review is the collection of the regulatory aspect of electrospun nanofibers as future medical devices. This part is added to the last paragraph of the introduction:
The healthcare industry is regulated by strict rules, so considering regulatory requirements at the development stage can be worthwhile. To the best of the authors' knowledge, there is no published review of the regulatory aspects of nanofibers developed as potential medical devices in the literature. In the first part of this review, the basic principles of electrospinning, the effects of the different parameters on the process and the produced fibers, and the types of solvent and melt electrospinning techniques are discussed. In the second part, the focus is on electrospun medical devices. After a summary of the possible areas of applications, their regulatory approach is described.
- Some tables or figures are recommended to be added to summarize the electrospinning-related parameters, which could make the manuscript more readable.
Thank you for this suggestion. There is one Ishikawa diagram-like figure, on which all the important parameters are collected.
Figure 4. The most important parameters affecting the feasibility of nanofiber production by the electrospinning method.
- Some statements feel they are lacking references. Admittedly, some of these statement might be considered well known facts, the concepts mentioned might have been referenced previously or will be in the future, but it might still be pertinent add references next to these statements for new readers to the field, skim readers or people who don't necessarily want to go looking for the relevant reference.
Thank you. References have been checked and corrected.
- The authors should make a critical review instead of plain text flow. This means that a comparative discussion should take place assisted with categorization of the attributes of the different systems in each section/Tables.
Thank you for your comment. The manuscript can be divided to the following sections:
- summary about the principles of the electrospinning – mainly for “new readers”
- comparison of different ES methods (solution vs melt, nozzle-based vs nozzle-free, syringe-based vs syringe-free) – important background for session 4
- nanofibers as medical devices. Three main topic: wound dressings, tissue engineering scaffolds, face masks – examples and important background for session 4
- regulation of electrospun medical devices (comparison of the FDA and EMA definitions and rules, comparison of different ISO standards, thoughts to keep in mind during development)
Also, Table 1 collects some recent studies on electrospun nanofibers as potential medical devices, and Table 2 shows examples of commercially available electrospun medical devices.
Moreover, Figure 4 and Figure 5 can be used for the categorization of the related topics (ES and application areas of tissue engineering scaffolds).
- As well known that, electrospinning-based strategies have been widely investigated for biomedical application. Do they have any commercial products based on nanomaterial technique available now?
Thank you for the question. As Table 2 shows, there are some commercially available electrospun medical devices such as respiratory masks, stents, and wound dressings.
Table 2. Examples of commercially available electrospun medical devices
Brand name |
Intended use |
Approved |
Bio Hygienic Mask |
compostable mask with FFP2-like filtration capacity |
Spain |
Bioweb™ |
stent coating composite |
in the pipeline |
Cerafix® Dura Substitute |
regenerative dural repair patch |
USA |
Covora™ |
soft-tissue engineering matrix |
USA |
EktoTherix™ |
soft-tissue scaffold |
completed clinical trial |
Inofilter® 95/99 |
face mask |
USA |
PK Papyrus |
covered stent |
USA |
ReBOSSIS-J |
absorbent bone regenerated material |
Japan |
ReDura™ |
regenerative dural repair patch |
unknown status clinical trial |
Restrata® Wound Matrix |
absorbable wound dressing |
USA |
Rivelin® plain patches |
wound patches |
completed clinical trial |
- The challenges and future direction of electrospinning-based biomedical scaffolds should be discussed in the Conclusions section.
Thank you for the comment. It has been added to the text.
In the field of tissue engineering, the most investigated tissues and organs are the bones, the blood vessels, the skin, and other soft tissues (muscle, tendon, valve), but the research is also intensive on cardiac, neural, and cartilage replacements. The challenges and future direction of electrospinning-based biomedical scaffolds are the development of the capability for reproducible, industrial-scale production and extensive pre-clinical and clinical testing before commercialization [4].
- The paper contains some typo and graphical errors. Please read carefully and correct Them.
Thank you for drawing attention to this. The corrections are colored green in the manuscript.
We are very grateful for your suggestions and comments. Hopefully, the answers will be satisfying.
Best regards,
- Luca Éva Uhljar
prof. Rita Ambrus
- 01. 2023. Szeged, Hungary

This manuscript is a resubmission of an earlier submission. The following is a list of the peer review reports and author responses from that submission.
Round 1
Reviewer 1 Report
The review article entitled “Electrospinning of potential medical devices and their regulatory approach” is divided in two parts. The first part gives an overview of electrospinning process and the second part is focused on the use of electrospun nanofibers for biomedical applications. Finally, regulatory aspects are also overviewed. Although some parts of the article are very well written (especially the biomedical part), the electrospinning part is confusing and not well explained. Furthermore, it is not clear what is the novelty that this review offers compared to the already published ones. Therefore, this review does not offer sufficient new insights that would merit publication in pharmaceutics.
Additional comments:
- In this review solution and melt electrospinning are explained and claimed to be the only two electrospinning methods. In the last years “emulsion electrospinning” or “green electrospinning” has also been developed. The authors should also talk about “emulsion electrospinning” in the review.
- Figure 1 does not give any relevant information
- Electrospinning part is confusing and not properly explained. I would recommend first explaining solution electrospinning and then explaining the influence of the different parameters (as they mostly correspond to solution electrospinning) that affect in the electrospinning process (process parameters, materials properties and ambient parameters). Finally, I would explain melt electrospinning and I would recommend to clearly state the differences between solution and melt electrospinning.
- Parameters of Figure 3 do not match the ones mentioned in the text. There are more parameters in the figure than the ones explained in the text. I recommend to stick to the parameters explained in the text (as they are the most important ones) and removing the non-mentioned ones from the image.
- The polymer solution viscosity is related to the concentration and molecular weight of the polymer used. This should be explained in the text.
- Line 60: publishes (not publish)
- There is something wrong with the sentence in lines 80-83 is not grammatically or is confusing: “Second, for melt electrospinning, both the nanofiber-forming polymer and the active ingredient must be thermally stable, while some polymers (e.g., PP or PET) that cannot be dissolved are not processable by solution electrospinning.”
- Line 122: The most important process parameters are: The voltage, the tip-to-collector distance and the flow rate. Although the type of collector and its movement also affects the fibre morphology this parameter is not considered one of the most important ones
- Line 124: the most important materials properties are the viscosity (related to the molecular weight and concentration of the polymer), the conductivity, the surface tension and the volatility of the solvent.
- In line 132 it is said that the voltage has the highest influence on the morphology of the formed fibers. This is not correct. The solution viscosity is the parameter that has the strongest influence on final fiber morphology.
- Lines 157-158: “Since the average diameters were not significantly different it can be concluded that the distance has not affected it [18].” This is not correct. The tip to collector distance affects on the fiber morphology: Biomaterials 29 (2008) 1989e2006; Chem. Rev. 2019, 119, 5298−5415; Applied Surface Science 296 (2014) 221–230
- Lines 417 and 418: “For biosensors and other soft electronics, nanofibers have the advantages of flexibility and/or stretchability, conductivity, and transparency; as well as large surface area and diverse fiber morphology [73].” This sentence is not completely correct as not all the nanofibers are conductive or transparent…
- In section 4 “Nanofibers as medical devices” Table 1 is shown but the authors should comment a bit more on which are the requirement to use a nanofiber for a medical device. What types of polymer have to be used? Do they need to be compulsorily biocompatible? Do they need to be biodegradable? Do they have to have specific dimensions? Can we use any type of solvent to produce these nanofibers?
Author Response
Peer Reviewer 1
( ) I would not like to sign my review report
(x) I would like to sign my review report
English language and style
( ) English very difficult to understand/incomprehensible
( ) Extensive editing of English language and style required
( ) Moderate English changes required
(x) English language and style are fine/minor spell check required
( ) I don’t feel qualified to judge about the English language and style
Comments and Suggestions for Authors
The review article entitled “Electrospinning of potential medical devices and their regulatory approach” is divided in two parts. The first part gives an overview of electrospinning process and the second part is focused on the use of electrospun nanofibers for biomedical applications. Finally, regulatory aspects are also overviewed. Although some parts of the article are very well written (especially the biomedical part), the electrospinning part is confusing and not well explained. Furthermore, it is not clear what is the novelty that this review offers compared to the already published ones. Therefore, this review does not offer sufficient new insights that would merit publication in pharmaceutics.
Thank you for your comments and questions. The English has been corrected by a linguistic proofreader. The novelty of the review is the collection of the regulatory aspect of electrospun nanofibers as future medical devices. This part has been added to the last paragraph of the introduction:
The healthcare industry is regulated by strict rules, so taking regulatory requirements into consideration at the development stage can be worthwhile. To the best of the authors' knowledge, there is no published review of the regulatory aspects of nanofibers developed as potential medical devices in the literature. In the first part of this review, the basic principles of electrospinning, the effects of the different parameters on the process and the produced fibers, and the types of solvent and melt electrospinning techniques are discussed. In the second part, the focus is on electrospun medical devices. After a brief summary of the possible areas of applications, their regulatory approach is described.
Additional comments:
- In this review solution and melt electrospinning are explained and claimed to be the only two electrospinning methods. In the last years “emulsion electrospinning” or “green electrospinning” has also been developed. The authors should also talk about “emulsion electrospinning” in the review.
Thank you for the suggestion. A new paragraph was added to the text about emulsion electrospinning.
2.3.2. Emulsion electrospinning
During emulsion electrospinning, homogenous mixtures of two or more immiscible liquids are electrospun. The method is very similar to single-needle solution electrospinning, but the obtained nanofibers have a core-shell structure [59]. The advantages of the core-shell structure are to protect the biologically active material located in the core and to ensure its controlled release. In addition, emulsion electrospinning allows the possibility to omit the coaxial needle, which simplifies the production. Nevertheless, it can be considered a green method, since water can be used in the continuous phase and organic solvents can be reduced or avoided [60]. Indeed, the avoidance of toxic solvents is particularly important for nanofibers intended for biomedical use.
- Figure 1 does not give any relevant information
Thank you for the comment. Figure 1 shows a schematic representation of electrospinning but emphasizes the possibility of charging the collector and grounding the needle. Moreover, providing a schematic figure of electrospinning at the beginning of the article seemed to be beneficial.
- Electrospinning part is confusing and not properly explained. I would recommend first explaining solution electrospinning and then explaining the influence of the different parameters (as they mostly correspond to solution electrospinning) that affect in the electrospinning process (process parameters, materials properties and ambient parameters). Finally, I would explain melt electrospinning and I would recommend to clearly state the differences between solution and melt electrospinning.
Thank you for the suggestion. A new paragraph was added and the order of the topics was changed according to the suggestion.
2.3. Electrospinning methods
As already mentioned above, solution, emulsion and melt electrospinning can be distinguished. The three methods follow the same main principles, although there are notable differences. The solution and the emulsion electrospinning are very similar regarding the equipment, the advantages, and the limitations. The main difference is the type of polymer liquid, as their names imply.
On the other hand, for melt electrospinning, heating is required to melt the polymer. For this reason, an additional heating component is needed, which causes a more complex electrospinning setup. Also, both the nanofiber-forming polymer and the active ingredient must be thermally stable, which limits the choice of materials. However, some polymers (e.g., PP or PET) cannot be dissolved, only melted before spinning. Moreover, the absence of solvent makes the melt method environmentally friendly. And finally, melt electrospinning is comparable to additive manufacturing techniques and thus can be utilized to create special three-dimensional electrospun nanofiber scaffolds for regenerative medicine applications [5].
- Parameters of Figure 3 do not match the ones mentioned in the text. There are more parameters in the figure than the ones explained in the text. I recommend to stick to the parameters explained in the text (as they are the most important ones) and removing the non-mentioned ones from the image.
Thank you for the comment. Figure 3 not only summarizes but also complements the text. The more important factors are described in the text, but for the purpose of completeness, minor influencing factors are also shown in the figure. It can also be seen as a kind of colorful, modern Ishikawa-diagram that tries to bring all the factors together.
- The polymer solution viscosity is related to the concentration and molecular weight of the polymer used. This should be explained in the text.
Thank you for the suggestion. The following modifications were done.
2.2. Effects of different electrospinning parameters
The factors that determine the properties of nanofibers are usually grouped into three categories (Figure 3):
- Process parameters (high voltage, flow rate, distance between the Taylor cone and the collector);
- Material properties (viscosity - related to the molecular weight and concentration of the polymer, surface tension, conductivity, and volatility of the solvent);
- Ambient parameters (temperature and humidity).
As the viscosity of the polymer solution is related to the concentration and molecular weight of the polymer used, both higher molecular weight and higher concentration can lead to increased viscosity of the fluid [30].
- Line 60: publishes (not publish)
Thank you for the correction.
- There is something wrong with the sentence in lines 80-83 is not grammatically or is confusing: “Second, for melt electrospinning, both the nanofiber-forming polymer and the active ingredient must be thermally stable, while some polymers (e.g., PP or PET) that cannot be dissolved are not processable by solution electrospinning.”
Thank you for the comment. The sentence has been revised (see above).
- Line 122: The most important process parameters are: The voltage, the tip-to-collector distance and the flow rate. Although the type of collector and its movement also affects the fibre morphology this parameter is not considered one of the most important ones.
- Line 124: the most important materials properties are the viscosity (related to the molecular weight and concentration of the polymer), the conductivity, the surface tension and the volatility of the solvent.
Thank you for the comments. The sentences were revised.
2.2. Effects of different electrospinning parameters
The factors that determine the properties of nanofibers are usually grouped into three categories (Figure 3):
- Process parameters (high voltage, flow rate, distance between the Taylor cone and the collector);
- Material properties (viscosity - related to the molecular weight and concentration of the polymer, surface tension, conductivity, and volatility of the solvent);
- Ambient parameters (temperature and humidity).
- In line 132 it is said that the voltage has the highest influence on the morphology of the formed fibers. This is not correct. The solution viscosity is the parameter that has the strongest influence on final fiber morphology.
Thank you! We were misleading with this sentence. It has been corrected.
Among the process parameters, the applied high voltage has the greatest influence on the mechanism of electrospinning, and thus the morphology of the formed fibers.
- Lines 157-158: “Since the average diameters were not significantly different it can be concluded that the distance has not affected it [18].” This is not correct. The tip to collector distance affects on the fiber morphology: Biomaterials 29 (2008) 1989e2006; Chem. Rev. 2019, 119, 5298−5415; Applied Surface Science 296 (2014) 221–230
Thank you for pointing this out. The paragraph has been revised.
Also, to obtain uniform nanofibers on the collector, complete drying of the jet, meaning evaporation of the solvent or solidification of the melted polymer, is required. Therefore, the jet requires an appropriate flight time in the air while whipping toward the collector. With a longer distance, the flight time increases, and thinner fibers will be formed. In this way, the flight distance of the jet (tip-to-collector distance in the case of nozzle electrospinning) effects on the fiber morphology [16-18]. However, with the increase of the distance, the strength of the electric field drastically decreases since it is inversely related to the square of the distance. Hence, the proper distance for each electrospinning depends on the voltage and other parameters, which usually ranges from 10 cm to 25 cm.
- Sill, T.J.; von Recum, H.A. Electrospinning: Applications in Drug Delivery and Tissue Engineering. Biomaterials 2008, 29, 1989–2006, doi:10.1016/j.biomaterials.2008.01.011.
- Rogina, A. Electrospinning Process: Versatile Preparation Method for Biodegradable and Natural Polymers and Biocomposite Systems Applied in Tissue Engineering and Drug Delivery. Applied Surface Science 2014, 296, 221–230, doi:10.1016/j.apsusc.2014.01.098.
- Xue, J.; Wu, T.; Dai, Y.; Xia, Y. Electrospinning and Electrospun Nanofibers: Methods, Materials, and Applications. Chem Rev 2019, 119, 5298–5415, doi:10.1021/acs.chemrev.8b00593.
- Lines 417 and 418: “For biosensors and other soft electronics, nanofibers have the advantages of flexibility and/or stretchability, conductivity, and transparency; as well as large surface area and diverse fiber morphology [73].” This sentence is not completely correct as not all the nanofibers are conductive or transparent…
Thank you for the comment. This statement refers only to biosensors. The sentence has been revised.
Nanofibers offer the benefits of flexibility and/or stretchability, conductivity, and transparency, as well as a large surface area and diverse fiber morphology for biosensors and other soft electronics [73].
- In section 4 “Nanofibers as medical devices” Table 1 is shown but the authors should comment a bit more on which are the requirement to use a nanofiber for a medical device. What types of polymer have to be used? Do they need to be compulsorily biocompatible? Do they need to be biodegradable? Do they have to have specific dimensions? Can we use any type of solvent to produce these nanofibers?
Thank you for the questions. Section 5 provides the answers and information about the regulatory requirements.
When developing nanofibers as a potential medical device, there are a few things that should be considered in the early stages of research that could be important from a regulatory perspective. First, the materials used must satisfy the safety requirements. It is recommended to choose a polymer that is considered safe and has been approved by the authorities. Biocompatible and biodegradable polymers are generally preferred. A good strategy could be the innovation of an already authorized device by electrospinning. Campbell et al. made nanofibers from an FDA-approved cyanoacrylate polymer for closing endonasal surgical defects and compared them with Adherus®, an FDA-approved common dural sealant [108]. Second, attention should be given to the toxic residue of the solvent used in the electrospinning process. It is advisable to analyze the residual solvent content and, if necessary, execute post-drying. The use of non-toxic solvents is preferable to aggressive and toxic ones like chloroform and HFIP. Latter also can be used if the residual solvent is proven to be below the level of acceptance. Another option can be melt electrospinning, since the solvent is not used in this technique. Third, both the electrospinning process and the equipment itself must be suitable for precisely controllable and reproducible production which can be demonstrated by validation. In this regard, nozzle-based electrospinning is better than nozzle-free because, in the latter, simultaneous jets lead to non-uniform fiber diameter [29]. Moreover, cellular and in vivo experiments may require nanofibers produced in a cleanroom environment. And finally, following the encouraging in vitro results, in vivo animal studies are crucial, especially for wound dressings and tissue scaffolds, where nanofibers will interact with living cells. Fortin et al. published promising in vitro results followed by negative in vivo results with electrospun tubular vascular conduits. In vitro, the conduits significantly reduced protein absorption and enhanced the adhesion, proliferation, and retention of endothelial cells seeded on the surface. Therefore, an end-to-end common carotid bypass was performed in ten sheep, although there was no improvement in endothelialization compared to the controls [153].
Additionally, information about the importance of the diameter has been added to section 3.
As in wound dressings and tissue engineering, the ability of nanofibers to produce aligned scaffolds capable of mimicking the extracellular matrix is exploited [74,75]. The fiber diameter, the pore size, and the alignment of the nanofibers are important to mimic the nano-sized features of human tissues. These features may play an active role in regulating cell activities such as orientation, migration, proliferation, and differentiation. Ferraris et al. published an article about the nanofiber topography and cell behavior [76].
- Ferraris, S.; Spriano, S.; Scalia, A.C.; Cochis, A.; Rimondini, L.; Cruz-Maya, I.; Guarino, V.; Varesano, A.; Vineis, C. Topographical and Biomechanical Guidance of Electrospun Fibers for Biomedical Applications. Polymers 2020, 12, 2896, doi:10.3390/polym12122896.
We appreciate your time, questions and suggestions, and hope that the answers are satisfying.
Best regards,
- Luca Éva Uhljar
prof. Rita Ambrus
- 12. 2022. Szeged, Hungary

Reviewer 2 Report
Dear authors
The article is interesting, well written, very complete, with pertinent references, and with scientific soundness, but is a topic that is fully discussed in the past with a great number of reviews that describes almost the same information as this paper. In this sense, can you explain more in detail, or emphasize further the novelty of the discussions of this review? what is the different input of this paper compared to the others published in the past? This should be stated in the last paragraph of the introduction.
Figures 1 to 3 are pertinent and correspond to the information of the paper but are not original, we have in literature many similar figures, can you propose innovative figures that other researchers can opt for the use of your figures instead of others that are practically the same?
Table 1 is pretty interesting because these electrospun systems are for biomedical applications, I supposed all of them were tested in biological tissue, and since you have a regulatory aspects section, I think is more interesting to know which tissues or cell lines have been tested. This information can replace the collectors-type column that does not add any interesting observation since both types of collectors such as rotatory and dynamic are reported in the same application and no further discussions on fiber orientation related to the application are observed.
Fiber diameters are not an interesting parameter to be discussed in biomedical applications?. In other words, as an interesting question, there is any relationship between an specific fiber diameter and biomedical application?
Please make sure that important remarks in figure 4 and table 1 are added to the conclusion section.
Congratulations on your hard work!!!!
Author Response
Peer Reviewer 2
(x) I would not like to sign my review report
( ) I would like to sign my review report
English language and style
( ) English very difficult to understand/incomprehensible
( ) Extensive editing of English language and style required
( ) Moderate English changes required
(x) English language and style are fine/minor spell check required
( ) I don't feel qualified to judge about the English language and style
Comments and Suggestions for Authors
Dear authors
The article is interesting, well written, very complete, with pertinent references, and with scientific soundness, but is a topic that is fully discussed in the past with a great number of reviews that describes almost the same information as this paper. In this sense, can you explain more in detail, or emphasize further the novelty of the discussions of this review? what is the different input of this paper compared to the others published in the past? This should be stated in the last paragraph of the introduction.
Thank you very much for your comments and questions. The novelty of the review is the collection of the regulatory aspect of electrospun nanofibers as future medical devices. This part has been added to the last paragraph of the introduction:
The healthcare industry is regulated by strict rules, so taking regulatory requirements into consideration at the development stage can be worthwhile. To the best of the authors' knowledge, there is no published review of the regulatory aspects of nanofibers developed as potential medical devices in the literature.
Figures 1 to 3 are pertinent and correspond to the information of the paper but are not original, we have in literature many similar figures, can you propose innovative figures that other researchers can opt for the use of your figures instead of others that are practically the same?
Thank you for this comment. Since electrospinning has been extensively discussed, the figures are similar to others in the literature. However, all figures were created entirely by us, and we have tried to provide as much information as possible.
Figure 1 shows a schematic representation of electrospinning but emphasizes the possibility of charging the collector and grounding the needle.
Figure 2 shows the relationship between surface tension and localized charges in relation to the droplet shape to complement the text.
Figure 3 summarizes and complements the text. We hope that it breaks up the length of the text a little and thus makes the article more accessible.
Figure 4 is a pie chart created by the authors to visualize the proportion of articles found in the literature.
Table 1 is pretty interesting because these electrospun systems are for biomedical applications, I supposed all of them were tested in biological tissue, and since you have a regulatory aspects section, I think is more interesting to know which tissues or cell lines have been tested. This information can replace the collectors-type column that does not add any interesting observation since both types of collectors such as rotatory and dynamic are reported in the same application and no further discussions on fiber orientation related to the application are observed.
Thank you for the suggestion, the column of “Cell line/animal” has been added to Table 1. Please find it in the corrected manuscript.
Fiber diameters are not an interesting parameter to be discussed in biomedical applications? In other words, as an interesting question, there is any relationship between a specific fiber diameter and biomedical application?
Thank you for the question, it is important, indeed. The information with a reference was added to the section 3.
As in wound dressings and tissue engineering, the ability of nanofibers to produce aligned scaffolds capable of mimicking the extracellular matrix is exploited [74,75]. The fiber diameter, the pore size, and the alignment of the nanofibers are important to mimic the nano-sized features of human tissues. These features may play an active role in regulating cell activities such as orientation, migration, proliferation, and differentiation. Ferraris et al. published an article about the nanofiber topography and cell behavior [76].
- Ferraris, S.; Spriano, S.; Scalia, A.C.; Cochis, A.; Rimondini, L.; Cruz-Maya, I.; Guarino, V.; Varesano, A.; Vineis, C. Topographical and Biomechanical Guidance of Electrospun Fibers for Biomedical Applications. Polymers 2020, 12, 2896, doi:10.3390/polym12122896.
Please make sure that important remarks in figure 4 and table 1 are added to the conclusion section.
Thank you for the suggestion. A new paragraph was added:
The potential biomedical applications of electrospun nanofibers differ from drug delivery systems to various medical devices such as filters, soft electronics, tissue engineering scaffolds, and wound dressings. Generally, the large surface area, the tailorable morphology, and the large range of polymers that can be used are the common advantages. Moreover, nanofibers have additional benefits in every field of application, for example, small pore size and flexibility as face masks or biocompatibility and similarity to the extracellular matrix as wound care and implantable devices. In the field of tissue engineering, the most investigated tissues and organs are the bones, the blood vessels, the skin and other soft tissues (muscle, tendon, valve), but the research is also intensive on cardiac, neural, and cartilage replacements.
Congratulations on your hard work!!!!
Thank you again for your time and suggestions. We hope that the answers are satisfying.
Best regards,
- Luca Éva Uhljar
prof. Rita Ambrus
- 12. 2022. Szeged, Hungary

Reviewer 3 Report
Electrospinning is presently a hot tool for many scientific applications, including pharmaceutics. During the past three decades, numerous researches and also many reviews have been published. The present manuscript gives a conclusion on the applications of electrospun medicated nanofibers as medical devices. Although the topic is interesting, the manuscript is poor in organization.
1) The title doesn’t match the contents, “medical devices” is a big word for the contents, drug delivery systems, scaffolds, tissue engineering products, plant-------, too many medical devices!
2) The most recent developments of electrospinning are not included, let alone the new kinds of applications.
3) The limited Figures have a poor quality, Only 4 Figures, and Fig 1 to Fig 3 are all simple diagrams, which can be found here and there.
4) A kind review article should provide more useful opinions on the recent progresses, and also indicate the readers the future development directions. However, this manuscript piled up data, and from old literature.
Author Response
Peer Reviewer 3
( ) I would not like to sign my review report
(x) I would like to sign my review report
English language and style
( ) English very difficult to understand/incomprehensible
(x) Extensive editing of English language and style required
( ) Moderate English changes required
( ) English language and style are fine/minor spell check required
( ) I don't feel qualified to judge about the English language and style
Comments and Suggestions for Authors
Electrospinning is presently a hot tool for many scientific applications, including pharmaceutics. During the past three decades, numerous researches and also many reviews have been published. The present manuscript gives a conclusion on the applications of electrospun medicated nanofibers as medical devices. Although the topic is interesting, the manuscript is poor in organization.
Thank you for your comments and questions. The English has been corrected by a linguistic proofreader. The corrections can be found in the manuscript uploaded.
1) The title doesn’t match the contents, “medical devices” is a big word for the contents, drug delivery systems, scaffolds, tissue engineering products, plant-------, too many medical devices!
Thank you for the comment. “Medical devices” is indeed a broad category that covers a wide range of devices. This article focused specifically on wound dressings, scaffolds and face masks. The title has been rewritten accordingly:
“Electrospinning of potential medical devices (wound dressings, tissue engineering scaffolds, face masks) and their regulatory approach”
2) The most recent developments of electrospinning are not included, let alone the new kinds of applications.
Thank you for the comment. The novelty of the review is the collection of the regulatory aspect of electrospun nanofibers as future medical devices. However, non-conventional ES methods are mentioned in the first table (e.g., coronal ES, multi-nozzle ES, co-ES + electroblowing, coaxial ES, wet ES, bubble ES, near-field ES, melt ES writing, ES combined with extrusion-based 3D printing technology, sandwitch technology). All data gathered in Table 1. is not older than 4 years old.
3) The limited Figures have a poor quality, Only 4 Figures, and Fig 1 to Fig 3 are all simple diagrams, which can be found here and there.
Thank you for this comment. Since electrospinning has been extensively discussed, the figures are similar to others in the literature. However, all figures were created entirely by us, and we have tried to provide as much information as possible.
Figure 1 shows a schematic representation of electrospinning but emphasizes the possibility of charging the collector and grounding the needle.
Figure 2 shows the relationship between surface tension and localized charges in relation to the droplet shape to complement the text.
Figure 3 summarizes and complements the text. We hope that it breaks up the length of the text a little and thus makes the article more accessible.
Figure 4 is a pie chart created by the authors to visualize the proportion of articles found in the literature.
The resolution of the figures inserted in the manuscript meets the requirements of the journal.
Beside the figures, there is a large table containing data from 56 research articles about different electrospun future medical devices. The table contains the type of use, the ES method, the used polymers, solvents and the bioactive agent if it is relevant. The table has been extended with in vivo data, as well.
4) A kind review article should provide more useful opinions on the recent progresses, and also indicate the readers the future development directions. However, this manuscript piled up data, and from old literature.
Thank you for this comment. Most of the references are recent – published in 2019-2022. All the articles were published between 2018-2022 showed by Table 1.
The novelty in this paper is the regulatory section, which collects the latest regulatory guidelines. Also, the regulatory part is completed by the “Nanofibers as medical devices” section. On the other hand, the basics of ES at the beginning may be helpful for those who are not familiar with the subject. In this section about the general findings, there are references to older articles, indeed. However, this first section also contains recent information regarding e.g., nozzle-free ES or wet ES.
We would like to thank you again.
Best regards,
- Luca Éva Uhljar
prof. Rita Ambrus
- 12. 2022. Szeged, Hungary

Round 2
Reviewer 1 Report
Thank you for the revision. The paper has been significantly improved. Now, I recommend it for publication.
Reviewer 2 Report
Dear authors
In my point of view, the paper was improved significatively in order to be published in this important journal
just a last comment
I suppose it is quite difficult to prepare new figures in the time given for corrections, but still, figure 3 looks messy because of the parameters, is not possible to improve its esthetics?
Reviewer 3 Report
The authors have substantially improved the manuscript's quality. However, it can not be satisfied without even one Figure about the medical devices, the most important topic and also key word in the title! What can the readers of PHARMACEUTICS get from your review in an easy and effective manner? electrospinning? But even electrospinning contents are not state-of-art, overlooking most of the most important developments reported most recently!